# Personalized Use of an Adjustable Movement-Controlled Video Game in Obstetric Brachial Plexus Palsy during Physiotherapy Sessions at School: A Case Report

**DOI:** 10.3390/healthcare11142008

**Published:** 2023-07-12

**Authors:** Beatriz Domingo, Noelia Terroso, Martina Eckert

**Affiliations:** 1Department of Physical Therapy, CEIP Pinar de San José, 28054 Madrid, Spain; beadso@gmail.com (B.D.); noheliatg4@gmail.com (N.T.); 2Research Center for Software Technologies and Multimedia Systems for Sustainability (CITSEM), Universidad Politécnica de Madrid (UPM), Alan Touring St. 3, 28031 Madrid, Spain

**Keywords:** physical therapy, motivation, kinect exergame, obstetric brachial plexus palsy, upper limb function

## Abstract

This case study explores the use of a personalized, adjustable Kinect exergame in 10 physiotherapy sessions for a 10-year-old girl with incomplete right-sided obstetric brachial plexus palsy (OBPP). The aim was to observe the impact on the patient’s upper limb mobility that could be achieved through maximization of the player’s motivation, possibly due to continuous game parameter adjustments. It had been achieved that the patient was playing 87% of the total gaming time with a personally challenging setting that increased her arm speed from one to four movements. Strength in abduction and flexion were increased by 8 N and 7 N, respectively. Furthermore, the patient showed better muscular balance and an increase of 50% in speed of the Jebsen-Taylor hand function test (JTHFT). The patient reported high levels of motivation, low perception of fatigue, and just slight discomfort. The study found that the use of personalized video games as a complement to conventional physiotherapy can be successful in OBPP patients when the game allows for the adjustment of the difficulty level as a response to personal performance. Predefined difficulty levels and automatic performance analysis can be helpful. Results are promising; however, further research is needed to confirm the results.

## 1. Introduction

Obstetric brachial plexus palsy (OBPP) is a brachial plexus injury that can occur during childbirth due to the action of endogenous or exogenous forces or a traction mechanism. It is commonly caused by shoulder dystocia, but it can also result from conditions present before the onset of labor [1,2,3]. Incidence rates vary across studies, with estimates ranging between 0.1 and 6.3 per 1000 births [3,4,5,6].

The severity and prognosis of OBPP are typically evaluated between 3 and 9 months of age. In general, the spontaneous recovery rate is estimated to be between 66% and 82%, while around 18–34% present permanent deficiencies or some type of disability [3,6,7,8]. OBPP can be classified based on the location of the injury, the type of nerve involvement, or the distribution or branches of the injured plexus [4,5,9,10].

Complications resulting from denervation of the muscles in OBPP include atrophy and muscle imbalance, leading to sagittal and transverse plane shoulder imbalance and active restrictions in range of motion. Contractures may also occur, which can decrease range of motion due to changes in non-skeletal tissues and alterations in anatomical bone structures such as glenohumeral dysplasia, which is present in 58–74% of the cases studied with magnetic resonance imaging [6,8,11].

Surgical treatment for obstetric brachial plexus palsy (OBPP) involves primary nerve surgeries for children who do not show spontaneous recovery within the first three months of life. Secondary surgeries aim to correct bone and joint issues as well as muscle transpositions to minimize functional deficits and bone alterations [5,12].

Conservative treatment for OBPP focuses on preventing and treating muscle strength imbalances. Strategies that maintain or increase strength or create more balanced forces can help preserve motion and reduce deformity, ultimately increasing the use of the affected arm [8,11].

Palomo and Sánchez [13] conducted a systematic review of various physiotherapy techniques used to treat OBPP, including restriction-induced movement therapy, Kinesiotape^®^, electrotherapy, virtual reality (VR) with splints or orthoses, and the application of type A botulinum toxin. Only one of the reviewed studies [14] applied a VR software (Armeo^®^ spring, Hocoma AG, Volketswil, Switzerland, 2008) and showed that it was significantly more effective than the conventional physiotherapy program when used for improving upper extremity function in children under 10 years old with OBPP. A further study achieved significant improvements of the internal rotation of the shoulder using a Nintendo Wii game on 11 children with obstetric brachial plexus injury (OBPI) [15].

Virtual reality has been used as a flexible and effective tool in cognitive and motor rehabilitation for decades through video capture and video games (VG). They have been found to provide significant benefits for improving motor function, including upper extremities, hand coordination, balance function, gait, and postural control. Multiple systematic reviews have highlighted the effectiveness and clinical utility of VR or VG interventions controlled by movement. VR and motion capture devices have been shown to have beneficial effects in the rehabilitation of patients with neurological disorders, achieving high levels of compliance, motivation, and commitment that can help improve one or more levels of the International Classification of Functioning, Disability, and Health [15,16,17,18,19,20].

Video games, when combined with motor training for specific tasks, are considered an effective intervention for pathologies such as cerebral palsy [21] and OBPP [14]. Although few publications exist on the use of video games for OBPP, promising results have been shown in improving upper extremity functions with the use of VG with Armeo^®^ [13] and therapeutic video games with Kinect [22,23,24].

The basis for these benefits lies in the fact that video games offer challenging and fun environments that keep players motivated for longer periods of time, resulting in increased repetitions of a challenging functional task. This, in turn, favors cortical reorganization and produces neuroplastic changes in children that lead to an improvement in performance. Furthermore, directing attention to an external focus has been found to aid faster learning with greater precision of movement compared to requests from an internal focus [16,25,26,27].

Most studies are conducted using commercial video games that do not cater to the specific needs of patients who may not be fast or efficient enough to perform movements successfully. Therefore, reviews highlight the advantages of using systems and games designed specifically for certain populations. These systems have specific goals that present suitable challenges which adapt to the speed and quality of movement. They are adjusted at the beginning of the intervention and can be re-adjusted when the player improves to prevent boredom or frustration among participants [16,18,20,28,29].

Regarding the type of intervention, the use of video games (VGs) is suggested as a complementary therapy to conventional therapies and not as a substitute. The involvement of physical and functional rehabilitation professionals who accompany the patient is highlighted as an advantage, but the use of these devices is also proposed as home use programs that can help maximize the intervention [19,20]. This includes independent or interactive practice at home, with or without the supervision of a professional. Radtka et al. [28] suggest that training in a virtual environment should be followed by practice in the physical environment to ensure appropriate perceptual adjustment and the transfer of training benefits to activities of daily living performed in the real world.

Moreover, it is important to note that the reviews underscore the need for greater scientific support in future proposals and a higher number of studies that can evaluate whether the gains obtained are sustained and whether these improvements transfer to real-life tasks and activities [13,14,15,16,20].

Preliminary studies indicate that adjusting the game parameters is crucial for enhancing patient motivation and interest in playing, thereby increasing the amount of playtime and the repetition of gestures [24].

This case study was a collaboration between the Physiotherapy Department of the Pinar de San José Primary School in Madrid and the Research Group on Acoustics and MultiMedia Applications (GAMMA) at Universidad Politécnica de Madrid. Its objective was to measure the impact of using an adjustable video game developed by GAMMA on a particular child affected by OBPP through adaptation of the game difficulty to the child’s specific capabilities and maintaining a motivational challenge throughout all sessions. Our hypothesis is that a continuous re-adjustment of the game’s difficulty level and, thus, responding instantaneously to the player’s changing abilities (i.e., due to improvement or fatigue) could maintain the challenge at an optimum level and finally lead to noticeable improvements in the physical state of the player.

## 2. Materials and Methods

### 2.1. Patient Information

The patient was a 10-year-old girl who suffers from right-sided obstetric brachial plexus palsy (OBPP). She exhibited signs of partial axonal injury in the C5 to C8-D1 myotomes. The girl was enrolled in a public school program for students with motor disabilities and receives physical therapy care twice a week, with each session lasting 45 min.

At 11 months of age, the patient underwent a latissimus dorsi transfer to improve glenohumeral abduction and external rotation. At 4 years old, a coracoplasty was performed, followed by an adhesion release, a pectoralis major fasciotomy, and a subscapularis tenotomy plus acromioplasty four years later. Before the coracoplasty surgery, botulinum toxin was injected into the subscapularis. The patient’s most recent magnetic resonance imaging (MRI) showed a shoulder deformity resulting from grade III OBPP, with posterior subluxation of the humeral head. She does not use any type of orthesis.

### 2.2. Clinical Findings

During our physical examination (Adams test), we noted good alignment of the spine at the postural level, with slight unevenness of the scapulae and asymmetry of the flanks, but without pelvic unevenness. In terms of active shoulder movement, abduction was measured at 80°, internal rotation to the gluteus, and external rotation to the ear with forearm drive. Passive ranges of motion included an abduction of 130°, full external rotation with 90° of abduction, and internal rotation of 20°. Angles were measured manually using a goniometer. Based on the Medical Research Council (MRC) scale, a widely used, reliable and simple clinical assessment tool for grading muscle strength [30], the patient demonstrated muscular strength of 4+/5 for the biceps, 4/5 for the triceps, 3/5 for wrist extension, 4+/5 for wrist flexors, 4/5 for finger extension, and 4+/5 for finger flexors.

We observed that the patient exhibited a Mallet classification grade III pattern [31]. She experienced difficulties with daily activities, such as dressing, combing, and washing her hair, as well as with precise bimanual activities such as tying shoelaces. Difficulties were also noted with finger dissociation. The patient displayed good integration of the limbs and a functional hand, as well as good sensory-perceptual discrimination at both tactile and proprioceptive levels. For the assessment of proprioception, we used blind recognition of joint positions. We performed a haptic perception at the stereognosis level to recognized regular flat and three-dimensional figures, textures, and lines on the skin.

### 2.3. Environment of the Study

The game utilized in this study was “Phiby’s Adventures v1,” part of the Blexer (Blender Exergame) system, created by the GAMMA research group and has been extensively described in [23,24,32]. It requires a normal PC connected to a Microsoft Kinect X360 camera for motion capture. The game contains four types of upper limb exercises, with difficulty (objectives and time limit) that can be set by the therapist prior to the exercise through the Blexer-med environment (database and web interface). The exercises are distributed in equal number of appearances throughout a map, and the order of performance is decided by the player.

The four exercises are:Chop: This includes raising and lowering the affected arm, requiring maximum shoulder flexion, posture stability to keep the axe raised until it is loaded, and a certain speed to lower the arm. The time limit and number of logs to be cut can be adjusted.Climb: This includes asymmetric movement in abduction of the arms and shoulder flexion, requiring an alternating movement of abduction. The time limit and height of the tree in meters (one meter is equivalent to one arm movement) can be adjusted.Row: This includes symmetrical movement of the upper limbs. of the time limit and width of the river in meters (one meter is equivalent to one back and forth arm movement) can be adjusted.Dive: This includes control and stability of the trunk. The time limit and number of plankton balls to be caught can be adjusted.

The game also features a motion amplification algorithm that provides equal experiences for players with different strengths and ranges of movement.

In a former study, we tested five adjustment levels of the game [24]:Very easy: ≤30% of the time limit used to achieve the objective.Easy: 30–70% of the time limit used to achieve the objective.Medium: 70–100% of the time limit used; 70–100% of the objective achieved.Hard: time limit reached; 30–70% of the objective achieved.Very hard: time limit reached; ≤30% of the objective achieved.

The outcome of the study was that the optimal settings for a player to maximize their motivation were “easy” and “medium” settings, according to their personal abilities. In these settings, the player is kept in a constant challenge (or flow), i.e., the player is moving close to the time limit and obtaining the objective but avoiding achieving both too easily. This avoids frustration or boredom. In the present case study, the difficulty has been re-adjusted at the beginning of each session in this sense to maintain the challenge of the game always at a maximum. The parameter adjustments and results obtained during play are visualized graphically and in the form of tables on the web interface, highlighted in red if the settings are too hard to make re-adjustments easier. Additionally, the web informs with special warnings if the player is performing in a hard or very hard setting.

### 2.4. Therapeutic Intervention

Figure 1 shows the timeline of the intervention. We started with two preliminary test sessions using the default settings, which presented a low challenge to allow the participant to become familiar with the activity. Afterward, the initial adjustments were made based on observations of the participant’s gameplay in that scenario. The intervention took place during the spring semester 2021/22, in 10 physiotherapy sessions scheduled during school hours, within a margin of six weeks (1 or 2 sessions per week). Test sessions and assessments were carried out before and after the 1st and after the 10th session. Each session started with the game; the duration of play was decision of the girl. For the remaining time of the session, conventional physical therapy treatment was performed. Outside school hours, the participant attended to her weekly swimming sessions.

The initial adjustments of time and objective was set to a combination of “easy” to “medium” difficulty, based on the student’s performance in the test sessions. Therefore, the time limit was set to 60 s for all exercises, and the objectives to be achieved for each of the four exercises were set as follows:Chop: 20 logs of woodClimb: 60 m (=60 alternate arm movements)Row: 30 m (=30 symmetric arm movements)Dive: 1 plankton

The game was played under the supervision of the physical therapist, who instructed and verbally motivated the girl. During each game session, the physiotherapist recorded the level of difficulty of the game to make the necessary changes to the parameter values to maintain a level of challenge that was adequate for the student’s abilities. Modifications were also made when severe fatigue was observed. The student was informed of the changes, and agreements were reached regarding a continuous increase in difficulty.

During the final part of the study, different activities were developed together with classmates to promote socialization, participation, and motivation regarding the intervention. We scheduled some sessions where the girl explained the video game to her classmates; they visited the physiotherapy room and played the game, and the person in charge of developing the video game at the University visited the class so the pupils could propose ideas for future game versions.

### 2.5. Assessment

The initial (v1) and final assessments (v2) were conducted using the following methods:Jebsen Taylor Hand Function Test (JTHFT): Improvement in this test is reflected by a decrease in the time used to complete each activity [33].Box and Block Test (BBT): Improvement is reflected in a greater number of blocks moved from one side of the box to the other [34].The strength of analytical movements of the affected limb’s (right arm) shoulder was measured using a Sauter model FA20 dynamometer.

Both the BBT and the JTHFT have shown reliability in measuring changes that focus on improving hand dexterity and function in children with CP after intensive interventions [33,34].

Additionally, at the end of each session, a follow-up questionnaire evaluated the subjective perception of fatigue (modified Borg scale), the presence of pain (visual analogue scale and pain location map), and motivation (visual tool with five faces) using the following questions:How tired do you feel?Is something hurting? Where? How much?How much did you enjoy the game today?Do you want to play again next week?

The questionnaires used have been uploaded as Appendix A.

## 3. Results and Outcome

It was possible to complete the 10 sessions in a maximum of six weeks, respecting the time that the student wished to play. The girl played one or two days a week, with a total playing time of 2 h and 46 min, which implies an average of 16 min per session, with a minimum time of 5 min and a maximum of 22 min.

### 3.1. Influence of the Game Difficulty Parameters

Throughout the 10 sessions, the challenge gradually decreased in all exercises except for Dive (Dive is a different type of exercise and difficult to control. When this scene was entered, the girl just waited for the timer to finish, which means that she was resting for 60 s). Therefore, the parameters had to be re-adjusted several times to adapt to the girl’s abilities and maintain a challenge and motivation (Figure 2). The Climb and Row exercise showed the most significant variation. The girl quickly learned and perfected the movements, and she was motivated by her increase in speed. In this way, she advanced in Climb from 20 movements in about 13 s (1.5 mov/s) on the first day to 150 movements in 46 s (3.3 mov/s) on the last day, which is a speed increase of more than double, considering that the exercise at the end lasted more than three times as long as at the beginning. In Row, she improved from 5 moves in 6 s (0.8 mov/s) to 150 moves in 40 s (3.75 mov/s).

Figure 3 shows an example graph provided by the Blexer-med platform. It illustrates the evolution of the Chop exercise with respect to the required time and the score achieved over time. Both are relative values (obtained value/objective). With some exceptions, it can be observed that the player reached the goal (blue line at 100%) using mostly 60% to 90% of the time limit. This corresponds to what we have defined as “medium” difficulty.

Through the adaptation of the difficulty parameters, it was achieved to maintain the challenge. The girl played in “easy” mode for about 49% of the total time, while for 38% of the time, the adjustment was in “medium” mode. Together, this corresponds to 87% of the total gaming time. The rest of the settings were “hard” (8%), “very easy” (3%), and “very hard” (2%) (Figure 4).

### 3.2. Assessment Results

In the following, we present the results of the initial (v1) and final (v2) evaluations.

The functional assessment reveals an increase in the speed of the JTHFT activities (Table 1). Speed increase ranged from 3% in lifting small objects to 50% in simulated feeding, with an average of 24%.

**Table 1 healthcare-11-02008-t001:** JTHFT initial (v1) and final assessment (v2).

Activity	v1 [s]	v2 [s]	v1–v2 [s]	Variation
Simulated feeding	22.64	11.35	11.29	50%
Lifting light objects	8.42	5.97	2.45	29%
Stack tokens	9.74	7.32	2.42	25%
Lifting heavy objects	11.26	8.5	2.76	25%
Turn pages	11.37	8.8	2.57	23%
Writing	80.0	69.96	10.04	13%
Lifting small objects	9.35	9.1	0.25	3%

The BBT assessment (Table 2) shows no variation between initial and final values.

Table 3 and Figure 5 present the measures of muscle strength of the right shoulder. An increase in strength was observed in the weakest muscle groups, which are responsible for abduction, flexion, and adduction. Abduction improved most. Other muscles that were initially strong slightly lost strength (extension, rotation). The average gain was higher than the loss and, overall, a better muscular balance was obtained (less deviation between muscle groups, orange circle in Figure 5) with a similar average strength. In the initial assessment, the maximum difference between forces was 13N, and after the intervention, it was reduced to 5N.

Table 4 presents the answers obtained in the daily follow-up questionnaire. The perception of fatigue was generally mild. Motivation was rated at maximum after each session, and the girl always stated that she wanted to play again. Pain was higher in the first sessions than at the end of the intervention. She reported daily discomfort that appeared during activity in different parts of the body, especially with intense activities, and subsided at the end with an average intensity of 2.6/10 (minimum 0–maximum 6). The discomfort was generally located in the costal and abdominal area. Only twice did she feel pain in the arm, located near the biceps.

## 4. Discussion

### 4.1. Influence of the Game Difficulty Parameters

An important aspect of this study was the analysis of the continuous adjustments made to the game parameters during the intervention and their influence on the development of the sessions. Comparing the parameter settings with the levels of motivation, pain, and fatigue, as well as the time the player spent in each exercise provides valuable insights.

As shown in Figure 4, the participant was able to play for most of the time (87%) at the difficulty levels that were initially deemed most suitable (“easy” and “medium”) due to the re-adjustments made before almost every session (Figure 2). This helped to avoid frustration or boredom and increased motivation, as evidenced by the participant’s responses to the questionnaire (Table 4). The participant experienced only mild levels of fatigue (2.2 out of 10) and slight discomfort that disappeared after the activity (2.6 out of 10). She maintained a consistently high level of motivation, expressing her desire to play again after each session. This promoted treatment adherence and increased the overall time of intervention in a motivating activity with repetitive functional gestures.

Reaching these results was possible due to the Blexer-med website, which provided a graphical visualization of the results (Figure 3) and additional advice in case the game was being played at a non-optimal level. This information helped to make modifications to the parameters based on concrete data rather than subjective evaluation and facilitated quick and instant re-adjustments of the challenge. Thus, the individual exercise difficulty could be adjusted on time to match instantaneously the new abilities the participant acquired through training, while maintaining a sense of challenge and the desire to overcome more difficult tasks. Additionally, the constant supervision and system warnings helped prevent the participant from overexerting her muscles, which could easily happen when children are over-motivated.

### 4.2. Assessment Results

Regarding the evaluation of possible motor benefits that could be obtained using a therapeutic video game, various positive aspects can be observed.

Firstly, evidence of a functional increase in simulated activities of daily living was found, as the speed of execution improved in all activities assessed with JTHFT (Table 1). This improvement in speed may be attributed to an increased use of the limb during play. The mandatory use of the affected limb to perform all the movements of the video game character could be considered a functional constraint-induced movement. In this case study, the student used her affected arm for an average of 16 min each session, intensely and in specific and repetitive tasks.

Moreover, the fact that the activity occurs with external attentional focus could improve the automation of the gesture, which subsequently influences the functional activities of daily life [27]. It is crucial to note that the intervention occurred in combination with the usual physiotherapy sessions and under the supervision of a physiotherapist. Therefore, the use of video games is considered complementary.

Secondly, an increase in the strength of certain muscle groups was achieved, which corresponds to the findings of El-Shamy and Alsharif’s [13] and Karas [15]. In the present study, a significant increase in the strength of the shoulder flexors and abductors was obtained. However, the most interesting finding, which differs from the findings of the other studies, was the improvement in muscular balance. After the intervention, there was greater harmony between the strength that the different muscle groups of the shoulder could develop (as seen in Table 3). This could help to preserve movements and control the deformity as the asymmetry of forces of the different muscle groups cause structural alterations [8,11].

Finally, in this case study, the possibility of carrying out activities with other classmates was a driving force to favor the inclusion of the student in the peer group. Her activities helped promote the exchange of experiences, common interests, and activities that positively reinforced her. According to the tutor’s interviews, this helped improve the student’s participation and inclusion in her group.

### 4.3. Strengths and Limitations

It is widely proven that the use of video games for therapeutic purposes achieves very high levels of motivation with hardly any secondary effects [35,36,37]. Nevertheless, commercially available games are generally not configurable, or the possibilities they provide are not sufficient for a precise adjustment to the personal needs and abilities of each patient. One of the strengths of using a configurable therapeutic video game like the one applied in this case study is the possibility of adjusting the parameters exactly to the personal requirements of the player and, furthermore, to maintain the challenge of the game in every moment. In this case study, adjustment criteria proposed in our previous studies were followed [24]; however, further research with a larger number of participants is necessary to validate these criteria. Therefore, we encourage other authors, who have more possibilities than us to recruit a larger number of children affected with OBPP, to perform an extensive study using an adjustable video game.

## 5. Conclusions

This case report shows the effectiveness of using a configurable video game as a complementary therapeutic tool in school physiotherapy sessions for a 10-year-old child with OBPP. The possibility to adapt the game’s difficulty levels throughout the sessions proved to be effective to maintain the challenge and motivation of the game for the patient, who experienced minimal fatigue and discomfort.

Through the intervention, an improvement in strength was observed in shoulder flexion and abduction movements, and greater balance was noted between the shoulder muscle groups. There were also significant improvements in motor function. In addition, the game was found to be a useful tool for improving participation and socialization in the school environment.

However, further studies with a larger number of subjects are needed to confirm these findings. Future research should focus on developing therapeutic video game environments that allow for quick and easy monitoring of playing progress, enabling physiotherapists to adjust objectives and challenges appropriately to everyone’s current abilities. Furthermore, it would be highly interesting to apply an artificial intelligence algorithm to perform automatic adjustments and send warnings to the therapist in case of abnormalities.

## 6. Patents

The software used in this case study has been registered as intellectual property with the following entry numbers: Blexer-med web platform 16/2019/1687; Middleware Chiro 16/2019/1576; and Phiby’s Adventures 16/2019/871.

## Figures and Tables

**Figure 1 healthcare-11-02008-f001:**
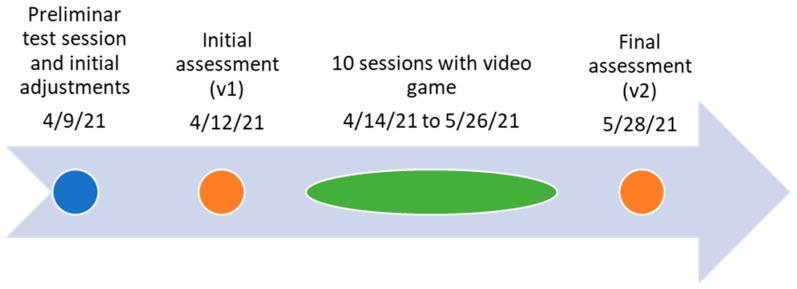
Intervention timeline.

**Figure 2 healthcare-11-02008-f002:**
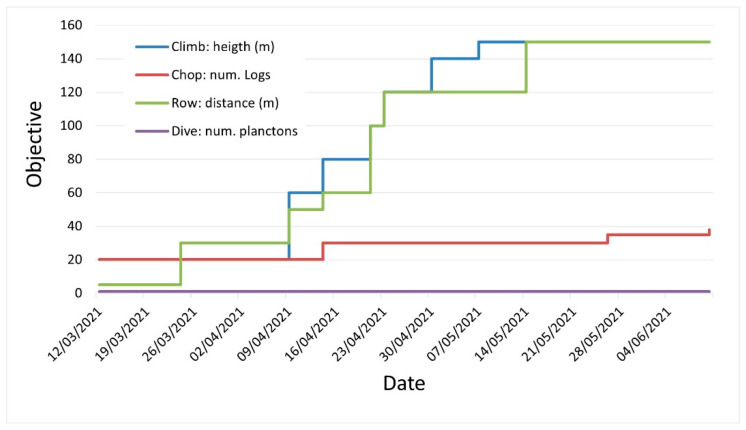
Re-adjustments of the objectives of each exercise throughout the intervention.

**Figure 3 healthcare-11-02008-f003:**
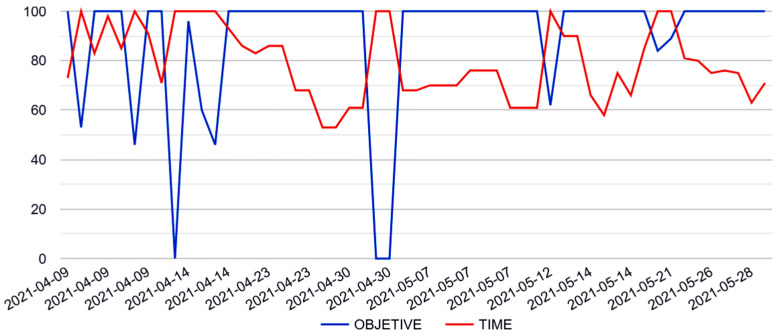
Visualization of the playing results on the Blexer-med web interface. In this graph, the percentage of the time limit used (red), and the percentage of the objective achieved (blue) can be observed.

**Figure 4 healthcare-11-02008-f004:**
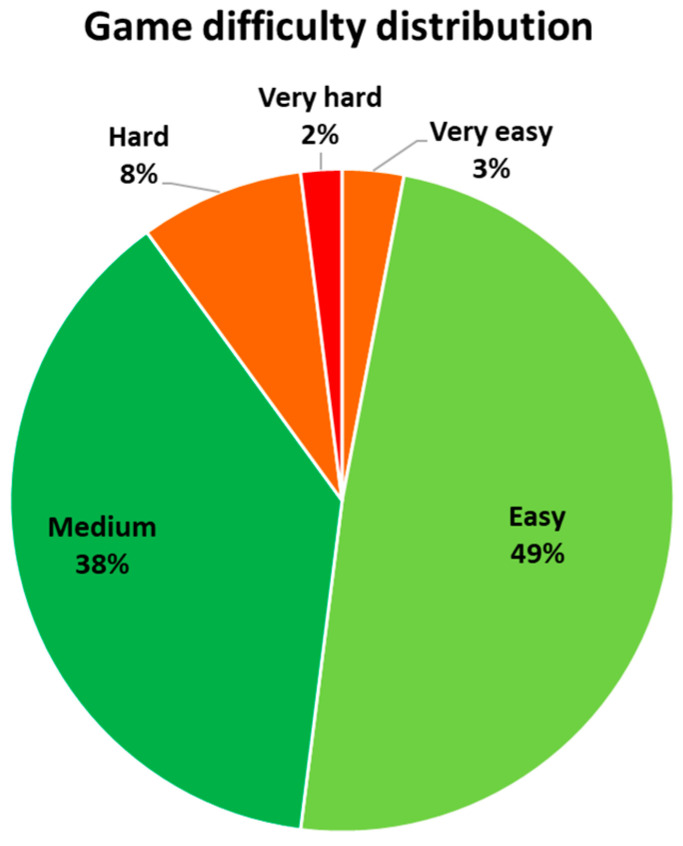
Percentages of time played in each difficulty level.

**Figure 5 healthcare-11-02008-f005:**
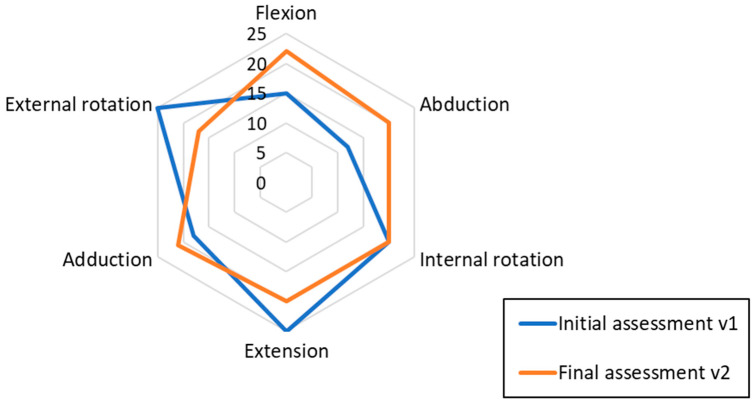
Muscular force measurements on the shoulder of the affected limb [N].

**Table 2 healthcare-11-02008-t002:** BBT initial (v1) and final assessment (v2).

Activity	v1 [n. Blocks]	v2 [n. Blocks]	v1–v2 [n. Blocks]	Variation
Move blocks	40	40	0	0%

**Table 3 healthcare-11-02008-t003:** Muscular force measurements on the shoulder of the affected limb (right arm).

Movement	v1 [N]	v2 [N]	v1–v2 [N]	Variation
Abduction	12	20	8	67%
Adduction	18	21	3	17%
Flexion	15	22	7	47%
Extension	25	20	−5	−20%
Internal rotation	20	20	0	0%
External rotation	25	17	−8	−32%
Mean (SD)	19.17 (5.27)	20 (1.67)	0.83 (6.43)	13% (38%)

**Table 4 healthcare-11-02008-t004:** Answers of pain monitoring (Borg Scale).

Date	Fatigue (1–10)	Motivation (1–5)	Play Again?	How Much Pain? (0–10)	Pain Where? ^1^
9 April 2021	2	5	yes	2	right abdomen
14 April 2021	2	5	yes	4	left abdomen
21 April 2021	2	5	yes	4	abdominals
23 April 2021	3	5	yes	1	arm-shoulder
27 April 2021	5	5	yes	6	left rib
30 April 2021	4	5	yes	1	left rib
7 May 2021	1	5	yes	1	right arm
12 May 2021	1	5	yes	4	left arm
14 May 2021	2	5	yes	1	abdominals
19 May 2021	0	5	yes	2	ribs
Mean	2.2	5		2.6	

^1^ On all occasions the discomfort appeared during the activity, but at the end it disappeared.

## Data Availability

The data obtained as gaming results are a large dataset that can be made available by the corresponding author. All other data obtained by measurements are published in the tables of this article.

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
