# Peer review of "Personalized Use of an Adjustable Movement-Controlled Video Game in Obstetric Brachial Plexus Palsy during Physiotherapy Sessions at School: A Case Report"

_healthcare, 2023, doi:10.3390/healthcare11142008_

Round 1
Reviewer 1 Report
First of all, I thank you for the opportunity to review this interesting contribution. This study describes Personalized Use of an Adjustable Movement-Controlled Video Game in Obstetric Brachial Plexus Palsy during Physiotherapy Sessions at School, and assesses the the impact of using an adjustable video game developed by GAMMA on a particular child affected by OBPP, through adaptation of the game difficulty to the child’s pecific capabilities and maintaining a motivational challenge throughout all sessions. There is, however, issues that must be resolved before the study can be accepted for publication. Several comments and suggestions for the authors:
· When the study was conducted?
· Patient Information - no information about the parent's consent to participate in the sessions/ The Ethics Committee – can you add the consent number?
· Clinical Findings (physical examination) - describe the test procedure and measuring tools used
· Lines 143-145: „The patient displayed good integration of the limbs and a functional hand, as well as good sensory-perceptual discrimination at both tactile and proprioceptive levels.” - How sensory-perceptual discrimination and proprioceptive levels was tested?
· Lines 194-195: „The remaining time of the session, conventional physical therapy treatment was performed” - What was the standard physiotherapy like, and was nothing new added to the standard therapy during the game session?
· Lines 229-231: „Both the BBT and the JTHFT have shown reliability in measuring changes that focus on improving hand dexterity and function in children with CP after intensive interventions.” - add references
· Results and Outcome - Were there sessions that took place day after day?
· Lines 160-161: „It is widely proven that the use of video games for therapeutic purposes achieves very high levels of motivation with hardly any secondary effects.” add references.
Author Response
Dear reviewer,
thank you very much for your time and for giving us the possibility to improve our manuscript. In the following, we will reply to all your concerns:
- When the study was conducted?
R: The study was conducted during the spring semester of 2021/22. We added this information in the section 2.5 Therapeutic Intervention, lines 189-191: “The intervention took place during the spring semester 2021/22, in ten physiotherapy sessions scheduled during school hours, within a margin of six weeks.”
- Patient Information - no information about the parent's consent to participate in the sessions/ The Ethics Committee – can you add the consent number?
R: We fully understand your doubts. There is a paragraph at the end of the article named “Informed Consent Statement”, where we have expressed the following: “The child's legal representatives provided informed consent for the child's participation in this study. Written informed consent has been obtained from the child's legal representatives to publish this paper.” Furthermore, in the paragraph “Institutional Review Board Statement” we expressed: “The study was conducted in accordance with the Declaration of Helsinki and approved by the Ethics Committee of Universidad Politécnica de Madrid (31/01/2018).” At that moment, the university did not register that approval below a consent number, but we provided the report to the editor and he accepted it.
- Clinical Findings (physical examination) - describe the test procedure and measuring tools used
R: Thank you for this hint. We used the Adams test to see if there is scoliosis. For angles, we used a standard goniometer. We added this to the text.
- Lines 143-145: „The patient displayed good integration of the limbs and a functional hand, as well as good sensory-perceptual discrimination at both tactile and proprioceptive levels.” - How sensory-perceptual discrimination and proprioceptive levels was tested?
R: For the assessment of proprioception, we used blind recognition of joint positions. We performed a haptic perception at the stereognosis level, to recognise regular flat and three-dimensional figures, textures, and lines on the skin. This information is now also added to the text.
- Lines 194-195: „The remaining time of the session, conventional physical therapy treatment was performed” - What was the standard physiotherapy like, and was nothing new added to the standard therapy during the game session?
R: Standard therapy includes passive, assisted mobilizations, muscle strengthening work, eye-hand coordination exercises, and postural correction work. Nothing new was added.
- Lines 229-231: „Both the BBT and the JTHFT have shown reliability in measuring changes that focus on improving hand dexterity and function in children with CP after intensive interventions.” - add references
R: Thank you, we added the references to those findings at the end of that sentence, they are the same as the ones cited earlier.
- Results and Outcome - Were there sessions that took place day after day?
R: The sessions were distributed throughout the intervention period with an average of two per week. We added this to the text in section 2.5.
- Lines 160-161: „It is widely proven that the use of video games for therapeutic purposes achieves very high levels of motivation with hardly any secondary effects.” add references.
R: Thank you for observing the missing references. We added the following articles, which confirm these facts:
- Horne-Moyer, H.L.; Moyer, B.H.; Messer, D.C. et al. The Use of Electronic Games in Therapy: a Review with Clinical Implications. Curr Psychiatry Rep 2014, 16, 520. doi:10.1007/s11920-014-0520-6.
- Zayeni, D.; Raynaud, J.P.; Revet, A. Therapeutic and Preventive Use of Video Games in Child and Adolescent Psychiatry: A Systematic Review. Front Psychiatry 2020, 11:36. doi:10.3389/fpsyt.2020.00036. PMID: 32116851; PMCID: PMC7016332.
- Granic, I.; Lobel, A.; Engels, R.C. The benefits of playing video games. The American Psychologist 2014, 69(1):66-78. DOI: 10.1037/a0034857. PMID: 24295515.
Reviewer 2 Report
Dear authors,
Congratulations on your work. It's very well conducted and written and complies with the CARE guidelines, with just a few missing points that I suggest you could add/explore:
(1) It would be good for the reader to have a timeframe in Figure 1 and when each outcome was measured (assessments).
(2) Is there a possibility to explore the patient's perspective on the intervention? Did you focus on this aspect anyhow?
(3) What were the roles of the parents/caregivers/legal representatives in this?
Some general formatting issues:
- As far as I could explore, reference [25] isn't cited in the text.
- Reference [30] is cited after [31].
Author Response
Dear reviewer,
thank you very much for your time and for giving us the possibility to improve our manuscript. In the following, we will reply to all your concerns:
(1) It would be good for the reader to have a timeframe in Figure 1 and when each outcome was measured (assessments).
R: Thank you very much for this hint. We added the dates of the assessments to the timeline to make it clearer.
(2) Is there a possibility to explore the patient's perspective on the intervention? Did you focus on this aspect anyhow?
R: In the questionnaire passed at the end of the session, the girl expressed her desire to play in the next session every day. Furthermore, she expressed repeatedly that she was very excited about playing the game and wanted to present it to their classmates, what she did. She also wanted to meet the researchers from the university who programmed the game and so we organized a visit in class and answered all the questions the pupils had.
(3) What were the roles of the parents/caregivers/legal representatives in this?
R: As this study took place in a primary school during regular classes, the family was informed but did not participate in the sessions.
Some general formatting issues:
- As far as I could explore, reference [25] isn't cited in the text.
- Reference [30] is cited after [31].
R: Thank you very much for your observations, we corrected these issues.
Reviewer 3 Report
Thank you for study.
There are a few points that should be added in the Method section.
I think that it will be more understandable if photographs including the position of the case and screenshots are added during the application in the study.
Information on when the case last received treatment and whether or not he used orthosis can be added.
In the muscle strength evaluation part, since there is no clear explanation about the reason why the extension and external rotation muscle strength decrease, it can be explained by adding photographs containing the position of the case. According to the muscle strength evaluation table, muscle imbalance seems to have increased. In this case, it may be necessary to look at and interpret the child's working positions.
Author Response
Dear reviewer,
thank you very much for your time and for giving us the possibility to improve our manuscript. In the following, we will reply to all your concerns:
I think that it will be more understandable if photographs including the position of the case and screenshots are added during the application in the study.
R: Thank you very much for your observation. We would like to present photos, but we did not even take them, because in Spain, it is illegal to take photographs in educational centres.
Information on when the case last received treatment and whether or not he used orthosis can be added.
R: The intervention with the video game took place over six weeks and she did not use any type of orthosis, we add this information in section 2.1. The regular physiotherapy treatment went on in parallel, as we explained in section 2.5: “The remaining time of the session, conventional physical therapy treatment was performed. Outside school hours, the participant attended her weekly swimming sessions.”
In the muscle strength evaluation part, since there is no clear explanation about the reason why the extension and external rotation muscle strength decrease, it can be explained by adding photographs containing the position of the case. According to the muscle strength evaluation table, muscle imbalance seems to have increased. In this case, it may be necessary to look at and interpret the child's working positions.
R: It is not possible to take photographs of the minor. However, by decreasing external rotation and extension, which are the ones that initially present greater muscle strength and increasing the strength of the rest of the movements, a greater similarity is produced between the values ​​obtained in all shoulder movements, thus improving the muscle balance. We added a graphical representation of the data to visualize this effect in Figure 5. You can see there that the orange circle is much more even than the blue one which represents the initial measures.
Reviewer 4 Report
Although a very interesting experiment, I was immediately inclined to reject the paper as this type of research cannot be generalized. Despite arguing that a larger number of subjects is needed (line 381), the authors wrongly conclude that one single case report can show the potential effectiveness of the technique (line 371). Furthermore, nothing is discussed in this regard in Section 4.3, line 359.
It would have been more appropriate “recruiting” additional patients to put forward a better and solid hypothesis. I do understand the difficulty of such endeavor, but it is a requirement, especially when dealing with health studies.
Author Response
Dear reviewer,
thank you very much for your time and for giving us the possibility to improve our manuscript. In the following, we will reply to your concerns:
It would have been more appropriate “recruiting” additional patients to put forward a better and solid hypothesis. I do understand the difficulty of such endeavor, but it is a requirement, especially when dealing with health studies.
R: We agree on the need for a bigger number of studies and a more significant number of cases. Nevertheless, this study took part after a similar test on 6 children with CP from the same school, where this girl was also participating. Results are published in [24]. During that study, we got aware of the good improvements she made, so we started this individual case study to get more detailed results we think that it could be worth publishing, such that other authors, who have more possibilities to recruit a larger number of individuals than us, could be encouraged to perform a more extensive study on children with OBPP.
We added this information in section 4.3 and changed line 371 accordingly.
Round 2
Reviewer 4 Report
Thanks to the authors for considering reviewer's comments.